# Evaluation of the Two-Point Ultrasound-Guided Transversus Abdominis Plane Block for Laparoscopic Canine Ovariectomy

**DOI:** 10.3390/ani12243556

**Published:** 2022-12-15

**Authors:** Lorena Espadas-González, Jesús M. Usón-Casaús, Nieves Pastor-Sirvent, Massimo Santella, Javier Ezquerra-Calvo, Eva M. Pérez-Merino

**Affiliations:** 1Department of Animal Medicine, Veterinary Faculty, University of Extremadura, Avenida de la Universidad s/n, 10003 Cáceres, Spain; 2Veterinary Teaching Hospital, Veterinary Faculty, University of Extremadura, Avenida de la Universidad s/n, 10003 Cáceres, Spain

**Keywords:** two-injection-point TAP, transverse abdominal plane, nerve block, laparoscopic ovariectomy, multimodal anesthesia, analgesia, dog

## Abstract

**Simple Summary:**

The transversus abdominis plane (TAP) block is an anesthetic technique that involves the injection of a local anesthetic to block the branches of the thoracolumbar spinal nerves innervating the abdominal wall and peritoneum. TAP has gained popularity in abdominal procedures, such as ovariectomy, but its use in veterinary laparoscopy remains poorly described. Among the different approaches described for TAP performance, the two-injection-point TAP results in a larger blocked area; however, clinical data on its efficacy are lacking. Our hypothesis is that a two-injection-point TAP could reduce the dose of intraoperative inhalational anesthetic and postoperative pain in dogs undergoing laparoscopic ovariectomy (LapOV). A total of 52 bitches were enrolled and divided into two groups: 26 were assigned to inhalational anesthesia, and 26 were assigned to inhalational anesthesia combined with TAP block. The end-tidal concentration of isoflurane and postoperative pain were assessed at different time points. The combination of the TAP block with inhalational anesthesia for the LapOV significantly reduced the requirements of isoflurane during the ovarian resection. Bitches that received TAP showed lower postoperative pain and required less analgesia intra- and postoperatively. The two-injection-point TAP block is an easy and effective anesthetic technique to provide postoperative analgesia to dogs undergoing LapOV.

**Abstract:**

The transversus abdominis plane (TAP) block causes desensitization of the abdominal wall and peritoneum. Of all the approaches proposed to perform it, the two-injection-point TAP showed the best results in terms of the area reached by the anesthetic solution. However, to date, no clinical data exist. The aim of this study was to evaluate the intra- and postoperative analgesic efficacy of a two-injection-point TAP block in dogs undergoing laparoscopic ovariectomy. A total of 26 animals were assigned to receive general inhalation anesthesia (control group), and 26 dogs were assigned to general inhalation anesthesia combined with TAP block (TAP group). The ultrasound-guided TAP block was carried out with a subcostal and cranial-to-ilium injection per hemiabdomen. The end-tidal concentration of isoflurane (EtISO) was recorded at different moments during the surgery. Postoperative pain was assessed at different time points during the first 24 h after surgery. The control group required significantly higher EtISO concentration during the ovarian resection and showed higher postoperative pain scores than the TAP group. Fewer dogs in the TAP group required intra- or postoperative rescue analgesia. TAP block can be implemented to improve postoperative pain management after laparoscopy, reducing the dosage of the systemic drugs used and, hence, their possible side effects.

## 1. Introduction

Laparoscopic procedures in dogs reduce postoperative pain and surgical stress and improve postoperative recovery compared to traditional open approaches [1,2,3]. Despite the minimally invasive nature of these techniques, patients may experience discomfort associated with the surgery in addition to abdominal distension and CO_2_ insufflation [4]. To provide intra- and postoperative analgesia for surgical procedures of the abdomen, the use of the transversus abdominis plane (TAP) block has been increasing for human and veterinary patients [5,6].

The TAP block is a locoregional anesthetic technique that causes sensory blockade of the abdominal muscles, subcutaneous tissues, skin, mammary glands, and parietal peritoneum, providing analgesia to the ventral and lateral abdominal wall [7,8]. Local anesthetic is injected into the fascial plane between the obliquus internus abdominis and the transversus abdominis muscles, anesthetizing the afferent branches of the ventral thoracic and lumbar nerves innervating the abdominal wall [9,10]. The locoregional anesthesia technique prevents the transmission of nociceptive impulses and produces satisfactory analgesia, helping to reduce opioid requirements and general anesthetics required for an optimal surgical plane of anesthesia [11].

In human medicine, the TAP block has been used successfully for pain relief in a wide variety of laparoscopic procedures [12,13,14,15,16]. Although the studies are less widespread in veterinary medicine, single-injection TAP blocks for conventional [17,18] and laparoscopic ovariectomy (LapOV) [19] in dogs have been recently reported. Different ultrasound (US)-guided TAP block approaches have been described in canine cadavers [5,10,20,21]. One of the latest studies on cadavers described a two-injection-point TAP technique (two-point TAP block) that showed the best results in terms of the area reached by the anesthetic solution [21]. However, no clinical data are currently available for this approach.

Therefore, this clinical investigation sought to evaluate the intraoperative antinociception and postoperative analgesic efficacy of a multipoint injection technique for TAP blocks in dogs undergoing LapOV. Our hypothesis was that a two-point TAP block would reduce the dose of intraoperative inhalational anesthetic and the need for postoperative analgesia.

## 2. Materials and Methods

### 2.1. Animals and Study Groups

A prospective, blind randomized study was done on 52 intact bitches from a local animal shelter (Cáceres, Spain). The study was approved by the University of Extremadura (UEx) Animal Care and Use Committee (register number 168/2021, approved 15 December 2021). Written informed consent was obtained from the manager of the animal shelter.

Before surgery, dogs underwent a thorough clinical examination, complete blood count, and serum biochemistry profile to ensure they were healthy. The selection criteria for patient inclusion were American Society of Anesthesiologists (ASA) category 1 and without comorbidities. As the surgical technique consisted of ovariectomy, dogs undergoing ovariohysterectomy for clinical reasons were excluded. Being younger than 6 months, showing any sign of systemic disease, hematological alteration, or pregnancy were other reasons to be excluded.

In this study, the patients were randomly distributed into two groups, depending on the anesthetic protocol: 26 dogs were assigned to receive general inhalation anesthesia (control group), and 26 dogs were assigned to general inhalation anesthesia combined with TAP block (TAP group). Each dog was assigned a number, and a random number generator was used to assign each patient to one of the two groups.

### 2.2. Surgical Procedure

All dogs included in this study underwent LapOV. The pneumoperitoneum was created using a Veress needle technique [22] and a mechanical insufflator set to 8–10 mmHg. To perform the three-port LapOV, two 5 mm portals were placed on the midline, in the middle and cranial position, while a 5 or 3 mm trocar was inserted in the caudal position depending on the animal’s size. A 5 mm 30° laparoscope (Hopkins II, Karl Storz Endoscopy, Tuttlingen, Germany) was used to visualize the abdominal cavity. The LigaSure vessel-sealing device was used to coagulate and cut the suspensory ligament, ovarian pedicle, and fallopian tube. After transection to remove the first ovary, it was grasped with laparoscopic Allis grasping forceps (Clickline, Karl Storz Endoscopy), and an attempt was made to remove it through the cannula. When the ovary exceeded the trocar size, the forceps were pulled backward until they entered inside the cannula, leaving the ovary stuck to the tip of the trocar. Then, the dog was rotated into the opposite recumbency, taking care of the forceps and ensuring that the ovary was attached. The free ovary was then released close to the second ovary while it was resected. Finally, both ovaries were removed from the abdomen sequentially, pulling out the middle or caudal cannula each time. The portal sites were sutured in two simple interrupted layers using 2/0 glyconate (Monosyn, Braun, Melsungen, Germany) for inner layers and 3/0 for intradermal sutures. A few cyanoacrylate tissue glue drops were left on each wound. The same specialized surgeons performed all the LapOVs.

### 2.3. Anesthetic Procedure

A pre-anesthetic evaluation was performed on every dog, including a physical examination, hematology, biochemistry, ECG, and two orthogonal thoracic radiographs.

All dogs were premedicated with intramuscular administration of methadone (Semfortan^®^ 10 mg/mL, Eurovet Animal Health BV, Bladel, The Netherlands) at 0.3 mg/kg and dexmedetomidine (Dexmopet, 0.5 mg/mL, Vetpharma Animal Health SL, Barcelona, Spain) at 3 mcg/kg. After a satisfactory level of sedation was obtained, one of the cephalic veins was catheterized for the administration of fluids (Ringer’s lactate solution 5 mL kg/h) and drugs. Anesthesia was induced with intravenous propofol (Propofol-Lipuro 10 mg/mL, B. Braun VetCare SA, Barcelona, Spain). Once the endotracheal tube was in place, isoflurane (IsoFlo, Zoetis Spain SL, Alcobendas, Madrid, Spain) in 100% oxygen was administered through a circular breathing system. During the surgical procedure in both groups, the anesthesiologist periodically assessed the absence of movement, jaw tone, palpebral reflex, and the ventromedial eyeball position to maintain an appropriate anesthetic plane, altering the vaporizer setting to increase or decrease the end-tidal isoflurane concentration (EtISO) by 0.5%. If the anesthetist observed signs of a sudden nociceptive response to surgery, a 0.5 mg/kg IV rescue bolus of propofol was administered.

Dogs were monitored using a multiparametric monitor (Carescape Monitor B650, GE Healthcare, Helsinki, Finland). Parameters recorded included heart rate (HR), end-tidal CO_2_, arterial hemoglobin saturation, the EtISO, temperature, tidal volume, compliance, cranial reflexes, and non-invasive blood pressure (oscillometric method). All dogs were mechanically ventilated using a pressure control mode to maintain end-tidal CO_2_ between 35 and 45 mmHg, with a maximum peak inspiratory pressure of 15 cm H_2_O and inspiratory:expiratory ratio of 1:3. All physiological data were recorded every 5 min.

In addition, prior to surgery, the dorsal metatarsal artery was percutaneously catheterized for invasive blood pressure monitoring. This catheter was removed before anesthesia recovery.

### 2.4. Tap Block Technique

Once the dogs were monitored and stable, they were placed in dorsal recumbency to perform the US-guided TAP technique. The hair on the abdomen was clipped, and the skin was aseptically prepared with chlorhexidine. The same experienced anesthetist performed the blocks using a 22-gauge, 88 mm or 22-gauge, 35 mm Quincke spinal needle (Spinocan, B. Braun, Recklinghausen, Germany). The anesthetist chose the needle size based on each dog’s size, body condition, and personal experience. The needle was connected to an extension set and a syringe (BD Discardit II, Becton Dickinson SA, Madrid, Spain). A linear 11.7 MHz US probe attached to a US machine (GE Logiq S7 Expert, GE Healthcare) was used in all the dogs.

The US-guided TAP block was carried out with two injections per hemiabdomen in each dog, following a modification of the technique described by Romano et al. [21]. The US transducer was firstly placed parallel to the caudal border of the last rib (subcostal approach) and, secondly, cranial to the crest of the ilium (umbilical approach) with a transverse orientation (Appendix A). Clear imaging of the three abdominal muscle layers (external abdominal oblique, internal abdominal oblique, and transversus abdominis) at both points were obtained. The needle was introduced in a cranial-to-caudal direction for the subcostal approach and in a ventral-to-dorsal direction for the umbilical approach. An “in-plane” technique was used to obtain a continuous real-time visualization of the needle until the tip reached the plane between the rectus abdominis and the transversus abdominis muscles for the subcostal approach and between the obliquus internus and transversus abdominis muscles for the umbilical approach (Appendix A). After the aspiration of blood produced a negative result, a small amount of the solution was injected into the virtual space to confirm the correct positioning of the needle. If the US imaging suggested that the location was incorrect and hydrodissection was not observed, the needle was repositioned, and the test injection was repeated. When the correct location was confirmed and the hydrodissection visualized, the remaining volume of the anesthetic solution was injected. A total volume of 0.3 mL/kg of 0.9% sodium chloride (B. Braun Vet Care, Milano, Italy), including a dose of 0.5 mg/kg of bupivacaine 0.5% (Bupivacaine 0.5%, B. Braun, Recklinghausen, Germany) was administered at each injection site. Surgery began 10 min after performing the anesthetic block. After the excision of the second ovary, a dose of IV meloxicam (Meloxidolor, 5 mL/mL, Le Vet Beheer BV, Oudewater, Netherlands) at 0.2 mg/kg was administered.

At the end of the surgery, isoflurane and fluid therapy were discontinued in both groups, and dogs were extubated when the swallowing reflex returned. All dogs were hospitalized for 24 h after surgery. During the postoperative period, buprenorphine (Buprecare, 0.3 mg/mL, Ecuphar NV, Oostkamp, Belgium) at a dose of 15 mcg/kg IV was administered as rescue analgesia if the evaluator recorded scores above five on the Glasgow Composite Pain Measurement Scale (CMPS), five on the Melbourne Pain Scale (MPS), or three on the Colorado Pain Scale (CPS). After 24 h, all the dogs were discharged.

### 2.5. Recorded Data

All study patients’ age, weights, and breeds were registered.

The EtISO (%), HR (beats/min), and mean invasive blood pressure (MAP; mmHg) were recorded at four different time points during the surgery: at the beginning of the surgery, before the pneumoperitoneum (Preop), after the resection of the first ovary (O1), after the resection of the second ovary (O2), and at surgical closure after the pneumoperitoneum deflation (Postop). The number of dogs that needed rescue intraoperative or postoperative analgesia in each group was also listed.

The total time to perform the four TAP blocks (from the moment of probe placement at the first point until the end of solution injection at the last injection point), the anesthetic time (from the intubation to the isoflurane administration ended), and the surgical time (between the Veress’ needle insertion and the end of the dermorrhaphy) were recorded.

Measurement of postoperative pain following LapOV with or without TAP block was performed at 2–3 (T1), 6–8 (T2), and 20–24 h (T3) after surgery using the CMPS, the MPS, and the CPS.

There was no sham or placebo injection in the control group, so the surgical team was not blinded to the randomization. The anesthesiologist that recorded the intra- and post-operatory variables was blinded to the treatment group and different from the anesthesiologist that induced the dogs and performed the TAPs.

### 2.6. Statistical Analysis

Data were analyzed using the Statistical Package for the Social Sciences (IBM SPSS for Windows, version 27.0; IBM Corp, Armonk, NY, USA). The Shapiro–Wilk test was applied to analyze the normal distribution of the data. Descriptive statistics of all study variables were analyzed. Age, weight, time to perform the four TAPs, anesthetic time, and surgical time were compared between the two groups of animals using Student’s *t*-test. Two-way repeated-measures ANOVA was used to evaluate changes in % EtISO, HR, and MAP over time and compare these variables among the two groups. Multiple pair-wise post hoc comparisons were made for significant fixed effects, and these comparisons were adjusted using Tukey honestly significant difference to avoid alpha risk inflation. The data were transformed, if necessary, by means of the natural logarithm or the square root to achieve normality. All those data are shown as means and standard deviations (SD). Regarding the pain scales, non-parametric tests (Mann–Whitney U) were used to compare both groups, and Friedman’s test was used to analyze the evolution within each group. The incidence of the need for rescue analgesia was compared with the χ^2^ test. The level of statistical significance was set at *p* < 0.05.

## 3. Results

Age and weight did not differ between groups. Ninety percent of the dogs in each group were greyhounds, and the rest were mixed breeds. The accomplishment of the four TAP block injections took an average of 12.25 ± 2.17 min. Anesthetic time was significantly longer for the TAP group than for the control group. No differences were found in the surgical time between the two groups. When comparing the number of dogs receiving intra- or postoperative rescue analgesia, no significant difference was observed (Table 1). No complications related to the anesthetic technique or the surgery were reported.

EtISO was significantly affected by time, treatment, and the interaction of treatment and time. EtISO concentration increased significantly at O1 and O2 time points in both groups. However, EtISO was significantly higher in the control group than in the TAP group during the time points corresponding to the ovarian resection (O1 and O2).

The HR was affected by time, increasing significantly at O1 and O2 in both groups. No differences were found in the HR between the two groups at any time point. There was no difference between the pre- and post-operatory EtISO and the HR in the TAP group, while the postoperative EtISO and HR of the control group remained significantly higher. No differences were observed in the MAP between groups or over time (Figure 1; Appendix A).

In both groups, the CMPS, MPS, and CPS pain scores increased significantly at T1. The CMPS and MPS pain scores at T1 did not differ from the preoperative scores in the TAP group. The CMPS and MPS pain scores at T1, T2, and T3, and the CPS scores at T1 and T2 were significantly higher in the control group than in the TAP group (Figure 2; Appendix A).

## 4. Discussion

The present study demonstrated that the combination of the TAP block with inhalational anesthesia significantly reduced the requirements of isoflurane needed to maintain the anesthetic plane suitable to perform the LapOV. This result contradicts one finding of the only study on TAP for LapOV, which did not find any difference in the median EtISO with their control group [19]. However, both studies show substantial differences in their design. For instance, while the TAP and control groups in the present study differ only in the TAP block, the comparison group in the study of Paolini et al. received a fentanyl-loading dose, followed by a constant rate infusion during the inhalation anesthesia [19]. The difference in the anesthetic protocol between the groups might have biased the results, as the authors themselves noted [19].

Our results completely agree with an earlier study on dogs undergoing conventional ovariohysterectomy that TAP block performed in the cranial and caudal abdomen with bupivacaine causes a significant reduction in isoflurane requirements [23]. The present study goes a step further, showing the reduction of the EtISO in the TAP group compared to the control group precisely during ovarian manipulation and resection, which has been proven to be the most painful stage of surgery in both open ovariectomies and LapOVs [6].

HR and MAP did not differ between the groups despite the different concentrations of isoflurane administered. The lower rise of isoflurane in the TAP group might be attributed to several reasons. One is the 20 min delay at the start of the surgery, although the anesthetic plane in the control group was considered proper when the surgery started approximately 5 min after the intubation. Moreover, the effect of the TAP block on visceral pain is controversial. Although traditionally, it has been considered that it does not achieve visceral analgesia [7], a clinical study described three cases of dogs with severe abdominal pain secondary to pancreatitis in two patients and laparotomy for the resection of a pancreatic tumor in a third dog, in which the pain was satisfactorily controlled thanks to the placement of catheters for performing a TAP block continuously. However, that was a single, non-comparative study including only three dogs [24]. The study on TAP for LapOV previously mentioned speculates that the blocks may have resulted in some degree of visceral analgesia [19]. On the contrary, a recent study assessing the analgesic efficacy of the TAP for canine ovariectomy did not observe any difference in the intraoperative variables measured between the group of dogs with TAP and the control group, supporting the hypothesis of the ineffectiveness of the TAP in blocking the visceral nociceptive stimulation [18]. Furthermore, the effectivity of the TAP block to desensitize the abdominal skin, subcutaneous tissues, abdominal muscles, and underlying peritoneum is demonstrated [7,10,19]. Thus, the painful stimuli due to stretching the peritoneal tissues and the abdominal wall and the peritoneal irritation after the CO_2_ insufflation could result in alleviation.

An additional advantage of the multipoint TAP technique of the present study is being less time-consuming. While a median of 25–26 min was required to perform the TAP and the intercostal blocks in the study by Paolini et al. [19], accomplishing the four TAP blocks of this study took a mean of 12 min. This time is close to the range of time between 48 and 120 s required to complete a single TAP block described in a study on cadavers [25]. The operator’s experience seems decisive in shortening the completion time [26]. However, the performance of the two-point TAP block significantly extends the anesthetic time, adding approximately 20 min to the time under inhaled anesthesia (12 min to complete the injections and 10 min to allow the spreading of the drug before the surgery).

The longer anesthetic exposure to isoflurane in the TAP group might be another reason for the lack of differences in HP and MAP between the two groups.

In contrast, the surgical time was shorter in the TAP group, probably because the episodes of nociception during the pause in forced anesthesia of the surgical procedure were fewer than in the control group, although without statistical significance. The surgical time in the present study (approximately 30 min) is considerably shorter than reported in the study that combined TAP with the intercostal blocks for LapOV, which was about 90 min, although the authors did not define the limits of the surgical time [19].

The two-point TAP injection has been previously proven to reach branches of nerves T13 to L3 [5] and has been successfully used for canine ovariohysterectomy [23] and feline ovariectomy [27] as well as canine mastectomy in combination with the serratus plane block [7]. LapOV requires two infraumbilical and one supraumbilical port. Thus, as used in the present study, a two-point TAP technique would be advisable to cover the area between the iliac crest and the 12th rib.

The TAP technique used in the present study is a mixture of the two approaches described in the study of Romano et al. on cadavers [21]. That study demonstrated that performing two blocks, one caudal to the costal arch and another lateral in a point between the last rib and iliac crest with the needle in a craniocaudal direction, resulted in a larger blocked area, staining nerves cranial to T12, compared with the approach that uses two injections in a ventrodorsal direction located caudal to the last rib and cranial to the iliac crest [21]. The broader area of desensitization of the abdominal wall and peritoneum achieved with this approach could be of interest to alleviate the discomfort due to the generalized stretching of the peritoneal tissues during the pneumoperitoneum. In this study, we performed the subcostal injection in the craniocaudal direction, while the injection cranial to the iliac crest was oriented from ventral to dorsal. The authors of the only other study on TAP applied to canine LapOV suggested this two-point TAP approach as a valid alternative to the combination of TAP with the intercostal block they used because it avoided clipping an additional area and the potential complications associated with the intercostal injections [19]. The present study confirms the clinical effectiveness of this approach.

As an anesthetic for TAP, bupivacaine has been shown to provide analgesia in dogs with abdominal pain [24] or undergoing ovariectomy [18] or mastectomy [28]. In a study on feline ovariohysterectomy, a mixture of bupivacaine and lidocaine was administered for the block [27]. Two studies on the performance of TAP for canine sterilization used dexmedetomidine [17] or ropivacaine [19] as the local anesthetic. However, no difference in the postoperative pain after the conventional ovariohysterectomy was observed using a mixture of bupivacaine and dexmedetomidine or bupivacaine liposome suspension for the TAP block [17]. Ropivacaine or bupivacaine have the same clinical uses at equivalent doses; however, ropivacaine is slightly less potent in the motor blockade and causes a shorter sensory blockade [29]. Both drugs administered intraperitoneally provided comparable postoperative analgesia in dogs after ovariohysterectomy [30], but the duration of the block of the brachial plexus with ropivacaine was shorter compared with that of the bupivacaine [31]. Despite this, no dog in this study or a similar study using ropivacaine [19] required postoperative rescue analgesia during the first 24 h post-surgery.

The present study found lower postoperative scores and less demand for rescue analgesia in the group of dogs in which TAP block was accomplished compared with dogs without TAP. Similar findings were reported in other studies using TAP to perform ovariectomy via celiotomy [17,18,24] or laparoscopy [19], according to the CMPS. The longer anesthetic time in the TAP group could have biased the results of the pain scores at T1. However, we did not observe differences in the time to extubation or to sternal recumbency between both groups. Therefore, it can be assumed that dogs from both groups were in similar conditions when the pain scores were recorded. The present study incorporated an additional two pain scales that bolstered those results. It has been reported that the signs associated with pain in dogs after soft tissue surgery decreased markedly 24 h postoperatively [32]. Therefore, the first 24 h post-surgery can be considered the critical period when pain should be closely monitored and alleviated. The preoperative administration of meloxicam has been shown to control pain effectively for 24 h after LapOV in dogs, but rescue analgesia might be required in some [33], as noted in the control group of the present study.

One of the limitations of the present study might be the lack of a control group receiving a TAP block with saline solution. Nowadays, the ethical management of animal shelters is under close and constant surveillance, and the sham treatment was rejected because of the additional unnecessary harm inflicted on the animals. Moreover, our aim was to recreate two scenarios that might frequently occur in daily clinical practice. The high number of greyhounds in our sample population might bias some results of the study, as their metabolic and anatomical features might be specific and different from other breeds. Another limitation is that the study design does not include a system to evaluate visceral pain, which would be especially desirable.

## 5. Conclusions

The modified two-point TAP block approach described by Romano et al. can be accomplished in a short time and without complications by experienced anesthetists. In the present study, the concentration of isoflurane required to maintain the anesthetic depth and the need for postoperative opioids in dogs undergoing LapOV with TAP block was lower than in the control group. According to these results, the TAP block can be implemented to improve postoperative pain management after laparoscopy, reducing the administration of the systemic drugs used and, hence, their possible side effects.

## Figures and Tables

**Figure 1 animals-12-03556-f001:**
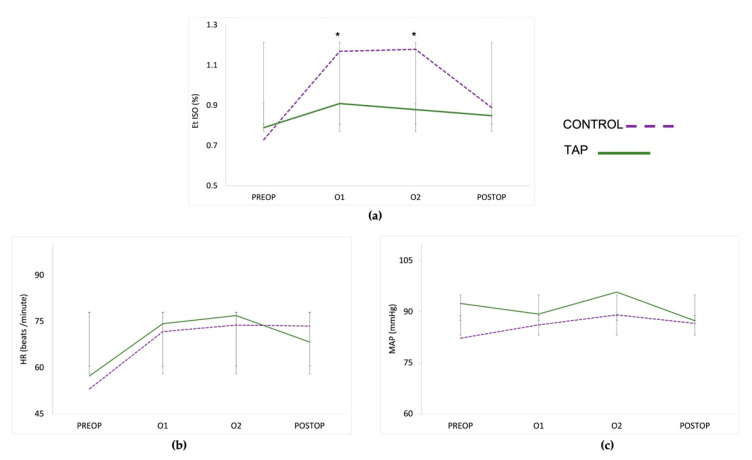
(**a**) Values of end-tidal isoflurane concentration (EtISO), (**b**) heart rate (HR), and (**c**) mean invasive blood pressure (MAP) at the beginning of the surgery (Preop), after the resection of the first ovary (O1), after the resection of the second ovary (O2), and at surgical closure after pneumoperitoneum deflation (Postop). The asterisk indicates a significant difference between the two groups at those time points. * *p* < 0.0001.

**Figure 2 animals-12-03556-f002:**
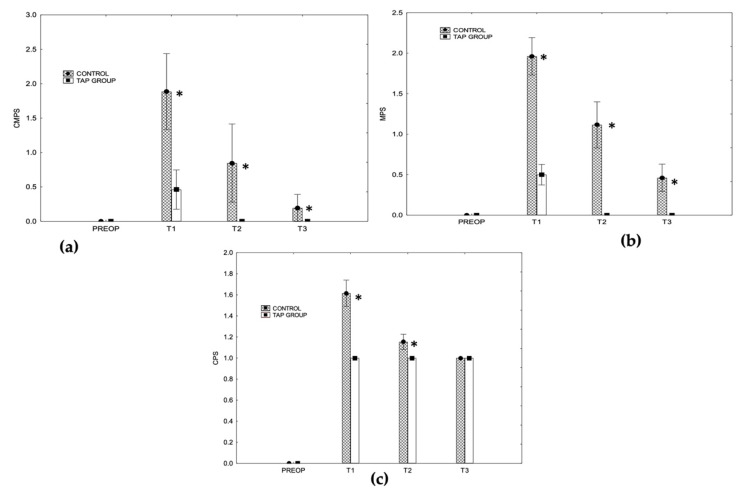
The bars indicate the pain scores’ mean and standard deviation values for both groups at the different time points according to the (**a**) Glasgow Composite Pain Measurement Scale (CMPS), (**b**) the Melbourne Pain Scale (MPS), and (**c**) the Colorado Pain Scale (CPS). The asterisks (*) indicate significant differences between both groups.

**Table 1 animals-12-03556-t001:** Comparative results of population and perioperative parameters between dogs that underwent laparoscopic ovariectomy with or without TAP block. Data are shown as mean ± SD.

	Control Group	TAP Group	*p*
Age (months)	27.85 ± 19.9	22.85 ± 14.43	0.32
Weight (Kg)	18.77 ± 6.03	19.05 ± 8.3	0.89
Anesthetic time (min)	41.23 ± 7.13	62.87 ± 4.13	0.001
Dogs requiring intraoperative rescue analgesia	7/26	3/26	0.30
Surgical time (min)	32.77 ± 6.49	29.85 ± 4.19	0.06
Dogs requiring postoperative rescue analgesia	2/26	0/26	0.71

## Data Availability

The data presented in this study are available on request from the corresponding author.

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
