# Peer review of "Evaluation of the Two-Point Ultrasound-Guided Transversus Abdominis Plane Block for Laparoscopic Canine Ovariectomy"

_animals, 2022, doi:10.3390/ani12243556_

Round 1

Reviewer 1 Report

thank's to the authors for this interesting paper which is well writed. Nevertheless, i have some questions for the authors before the acceptation. 

You say that the anesthetist was blind to the randomization but surgeon wasn’t, which is not clear. I don’t understand why the surgeon assist or are present during the TAP block and why the anesthetist who realized the injection (logicaly in charge of the anesthesia procedure is blind to the randomization) => this means that the anesthesiologist that realized the injection is systematically different that the one who recorded information during the surgery and in post op ? please clarify this point.

Did you find or observe a difference in wake up delay between both group ?

As the TAP group sleeps longer did you not think that the more important isoflurane ipregnation in TAP group may hava an effect on the EtISO during surgery (especially since you don't observe HR and MAP difference)  and in your pain score the first hours in post op  (as it may make more time to wake up with longer anesthesia and lead to pain score misinterpretation) ? You should discusse that in your discussion part. 

It also seem very weird that the HR and MAP do not differ between groups while Etiso differ significantly. You should precise (with more details) in material and methods the reasons that make you increase the isoflurane concentration in groups and not in the other while your monitoring informations were theoricaly the same. as HR and MAP is considere as sign of stress/pain or awakness and do not diff between groups... you may have observe awakness signs or movement or may it have been a bias with a non blind observator... Please discuss that in your discussion part. 

All information in the discussion part treating about comparison between dual site TAP and one site TAP should be deleted as you do not compared it in your paper.

Author Response

REVIEWER 1:

Thank you for your kind comment about the interest of this research. We are grateful for your careful review and your constructive and thoughtful comments. We have read your comments carefully and tried our best to address them one by one, and we feel that the manuscript is greatly improved as a result. Our responses to your precise comments are as follows:

thank's to the authors for this interesting paper which is well writed. Nevertheless, i have some questions for the authors before the acceptation. 

You say that the anesthetist was blind to the randomization but surgeon wasn’t, which is not clear. I don’t understand why the surgeon assist or are present during the TAP block and why the anesthetist who realized the injection (logicaly in charge of the anesthesia procedure is blind to the randomization) => this means that the anesthesiologist that realized the injection is systematically different that the one who recorded information during the surgery and in post op ? please clarify this point.

Thank you for pointing this out, that might be unclear in the text.

The pace of work for each dog was as follows: The anesthetist 1 induced, intubated, positioned the dog, placed the sensors, and, once the animal was stable, performed the TAP (in one group) or not (in the other). Then she left before the beginning of the surgery and anesthetist 2 came to the operating room, monitored the anesthesia during the surgery, the dog’s recovery, and assessed the pain over the following 24 hours. While anesthetist 1 was in the operating room, some members of the surgical team could be scrubbing (out of the operating room) or already wearing surgical attire and preparing surgical instruments, or some others were setting in the laparoscopic tower, acting as circulant personnel. The ideal would have been to start this process when the TAP had finished and the anesthetist 1 left the operating room, but that meant a delayed start of the surgery. The second option was to allow the surgeon’s assistants to get everything ready while anesthetist 1 is working and the surgeons would enter to the operating room when she left. However, although two surgeons with expertise in laparoscopy performed all the surgeries, the assistants were often students and they needed surveillance and guidance to assure they were properly attired, the instrument table properly set up, etc., and surgeons acted as professors as well.

To clarify this, we have specified in the text that the anesthesiologist that recorded the intra- and post-operatory variables was blinded to the treatment group and different from the anesthesiologist that induced the dogs and performed the TAPs

Did you find or observe a difference in wake up delay between both group ?

As the TAP group sleeps longer did you not think that the more important isoflurane ipregnation in TAP group may hava an effect on the EtISO during surgery (especially since you don't observe HR and MAP difference)  and in your pain score the first hours in post op  (as it may make more time to wake up with longer anesthesia and lead to pain score misinterpretation) ? You should discusse that in your discussion part. 

Thank you for bringing these issues to our attention. We agree that the impact of isoflurane tissue storage depends on the duration of anesthesia. However, we did not find any difference in the wake-up time between both groups. You are right that longer exposure to isoflurane might influence surgery and the first pain score. However, regarding the first pain scores, we did not observe differences in extubation time or in the time to attain sternal recumbency. Then, it could be inferred that the animals were in similar conditions when the pain scores were recorded.

We have included these accurate observations in the discussion part.

 It also seem very weird that the HR and MAP do not differ between groups while Etiso differ significantly. You should precise (with more details) in material and methods the reasons that make you increase the isoflurane concentration in groups and not in the other while your monitoring informations were theoricaly the same. as HR and MAP is considere as sign of stress/pain or awakness and do not diff between groups... you may have observe awakeness signs or movement or may it have been a bias with a non blind observator... Please discuss that in your discussion part. 

The plane anesthetic required for the surgery was the same for both groups and the adjustments of the vaporizer were allowed in both groups, not only in the control group. The isoflurane adjustment was based on the absence of movement, palpebral reflex, and the ventral rotation of the eyeball These criteria have been now incorporated in the methods section of the manuscript following your suggestion.

As you correctly pointed out, we did not observe any difference in HR and MAP between the groups in spite of the different concentrations of isoflurane administered. The lower rise of isoflurane in the TAP group could be attributed to the 20-minute delay in the start of the surgery, although the anesthetic plane in the control group was considered proper when the surgery started approximately 5 minutes after the intubation. Additionally, although traditionally it has been considered that TAP block does not achieve visceral analgesia, a study described the usefulness of this block within a multimodal analgesia protocol in patients with visceral pain. This clinical study (Freitab FAV et al., 2018) included three cases of dogs with severe abdominal pain secondary to pancreatitis in two patients and laparotomy for the resection of a pancreatic tumor in a third dog. In the three dogs, the pain was satisfactorily controlled thanks to the placement of catheters. for performing a TAP block continuously. Furthermore, the effectivity of the TAP block to desensitize the abdominal skin, subcutaneous tissues, abdominal muscles, and underlying peritoneum, is demonstrated. Thus, the pain due to the stretching of the peritoneal tissues and the abdominal wall and the peritoneal irritation after the CO2 insufflation could result in alleviation.

Moreover, an early study of dogs anesthetized with two different doses of isoflurane did not find differences in HR or MAP, although they used a higher concentration (1.5 or 2 times the MAC) (Floriano et al., 2016).

All information in the discussion part treating about comparison between dual site TAP and one site TAP should be deleted as you do not compared it in your paper.

All that information has been removed following your indication.

Reviewer 2 Report

Thank you for submission  very interesting research paper that focus on the  regional nerve blocks. This is a very useful study that is a great deal of attention in clinical practice.

The information you gave us here will provide useful information for many clinicians.

 I do not see any major problems with the design of this study, but there are a few questions about your results and discussions.

I will address attached files.

Please answer them.

Author Response

We have expected your review and your questions but we have not received anything.

Reviewer 3 Report

This manuscript contains interesting information in veterinary practice, especially intraoperative and postoperative pain management. Since there is no evidence that TAP alleviates visceral pain, I would appreciate more information. Comments are included below.

Major Comments

l  A TAP block is a block of sensory nerves in the abdominal wall and has no effect on visceral pain. Despite this, what is the reason for the difference in inhaled isoflurane concentration between groups during ovarian removal, which causes visceral pain? More information is needed.

l  The difference in postoperative pain scores is only about 1 on the pain scale used in this study. Specifically, what category did you see a difference in? Are pain evaluators the same over time? Did you do it with blinds? How clinically meaningful are differences in pain scales?

Minor Comments

l  Lines 125-129. Adjusting the isoflurane vaporizer setting in 0.5% increments for proper adjustment of the depth of anesthesia is far too much. Why did you choose rescue with propofol, which has no analgesic effect, to respond to acute symptoms caused by nociceptive stimuli? The use of propofol and isoflurane introduces bias in adjusting the depth of anesthesia. Since TAP blocker is used for analgesia, wouldn't it be better to keep isoflurane levels constant and assess the frequency and amount of analgesics such as fentanyl?

l  Line 151-166. Please specify any reference references for the TAP block method.

l  Line 154-157. In this TAP block, did you administer the local anesthetic more laterally (dorsally) than the rectus abdominis is involved? Please explain in detail.

l  Line 156-157. Two TAP block approaches are important in this study. Please describe in detail in the supplementary file using a schematic diagram of the administration site and an ultrasound image of the muscle layer.

l  Line 194-196. Were postoperative pain assessments performed by an anesthesiologist? Did they assess pain blindly? Were they assessed consistently by the same person?

l  Line 219-220. "although it was higher in the control group." should be deleted. The results should be stated indifferently that there was a significant difference.

l  Figure 1. Is this an average value? If so, add standard deviation error bars. Also, is the increase in heart rate at O1 and O2 in both groups due to pain? Please consider.

Author Response

Thank you for your careful review and your kind comment about the interest of this research. We believe that your suggestions have been highly constructive. We have gone through your comments carefully and tried our best to address them one by one. We hope the manuscript has been improved accordingly. Our responses to your accurate questions are the following:

This manuscript contains interesting information in veterinary practice, especially intraoperative and postoperative pain management. Since there is no evidence that TAP alleviates visceral pain, I would appreciate more information. Comments are included below.

Major Comments

l  A TAP block is a block of sensory nerves in the abdominal wall and has no effect on visceral pain. Despite this, what is the reason for the difference in inhaled isoflurane concentration between groups during ovarian removal, which causes visceral pain? More information is needed. 

Although traditionally it has been considered that TAP block does not achieve visceral analgesia, a study described the usefulness of this block within a multimodal analgesia protocol in patients with visceral pain. This clinical study (Freitab FAV et al., 2018) included three cases of dogs with severe abdominal pain secondary to pancreatitis in two patients and laparotomy for the resection of a pancreatic tumor in a third dog. In the three dogs, the pain was satisfactorily controlled thanks to the placement of catheters. for performing a TAP block continuously. Furthermore, the effectivity of the TAP block to desensitize the abdominal skin, subcutaneous tissues, abdominal muscles, and underlying peritoneum, is demonstrated. Thus, the pain due to the stretching of the peritoneal tissues and the abdominal wall and the peritoneal irritation after the CO2 insufflation could result in alleviation.

The longer exposure to the inhaled anesthesia in the TAP group and the 20-minute delay at the start of the surgery might also have some impact, although the anesthetic plane in the control group was considered proper when the surgery started approximately 5 minutes after the intubation.

l  The difference in postoperative pain scores is only about 1 on the pain scale used in this study. Specifically, what category did you see a difference in? Are pain evaluators the same over time? Did you do it with blinds? How clinically meaningful are differences in pain scales?

 In the revised manuscript we have included a sentence clarifying that the anesthesiologist that performed the TAP and the one that monitored the anesthesia and postoperative pain were different. The second anesthesiologist was who assessed blindly the postoperative pain of all the dogs in this study. Differences were observed in activity regarding the Melbourne scale (at rest-eating-pacing) behavior concerning the Colorado and Glasgow scales (comfortable-distracted). As you accurately suggest, there is no clinical significance in these differences and they often depend on the dogs’ temperament.

Minor Comments

l  Lines 125-129. Adjusting the isoflurane vaporizer setting in 0.5% increments for proper adjustment of the depth of anesthesia is far too much. Why did you choose rescue with propofol, which has no analgesic effect, to respond to acute symptoms caused by nociceptive stimuli? The use of propofol and isoflurane introduces bias in adjusting the depth of anesthesia. Since TAP blocker is used for analgesia, wouldn't it be better to keep isoflurane levels constant and assess the frequency and amount of analgesics such as fentanyl?

Different studies on multimodal anesthesia used intraoperative propofol bolus in response to nociceptive response to surgery. (Lardone E. et al, BMC Vet Res. 2022;18(1):200; Gomes VH et al. Vet Anaesth Analg. 2020;47(6):803-809; Krohm P, et al. Vet Anaesth Analg. 2011;38(4):363-373). In those studies, as well as in the present study, propofol was chosen due to its short onset and duration of action while it does not provide additional analgesia. Therefore, it could be used to quickly deepen hypnosis without significantly masking the potential benefits of successful loco-regional anesthesia. The use of fentanyl could also bias the amount of isoflurane administered since the use of fentanyl allows the reduction of the MAC of the isoflurane in a significant percentage according to the dose used. Besides, fentanyl might result in bradycardia and hypotension after the administration, which could also bias those results.

Concerning the 0.5% increments, we apologize for not having explained ourselves properly. We meant to say that to maintain an appropriate anesthetic plane, the vaporizer setting was varied to increase or decrease the end-tidal isoflurane concentration (EtISO) by 0.5%. We have rewritten the paragraph to make that clear.

l  Line 151-166. Please specify any reference references for the TAP block method.

We performed the TAP based on the technique described by Romano et al. but with some modifications. So, we have cited that reference in the method.

l  Line 154-157. In this TAP block, did you administer the local anesthetic more laterally (dorsally) than the rectus abdominis is involved? Please explain in detail.

We have improved the explanation of the technique in the text, clarifying that for the subcostal approach the needle is inserted between the rectus abdominis and the transversus abdominis muscles, and between the obliquus internus and transversus abdominis muscles for the umbilical approach.

l  Line 156-157. Two TAP block approaches are important in this study. Please describe in detail in the supplementary file using a schematic diagram of the administration site and an ultrasound image of the muscle layer.

We have uploaded a supplementary file with ultrasound images and a diagram of the administration of the TAP block.

l  Line 194-196. Were postoperative pain assessments performed by an anesthesiologist? Did they assess pain blindly? Were they assessed consistently by the same person?

The answer to the three questions is yes. In the revised manuscript we have included a sentence clarifying that the anesthesiologist that performed the TAP and the one that monitored the anesthesia and postoperative pain were different. The second anesthesiologist assessed blindly the postoperative pain of all the dogs in this study.

l  Line 219-220. "although it was higher in the control group." should be deleted. The results should be stated indifferently that there was a significant difference.

The sentence has been deleted.

l  Figure 1. Is this an average value? If so, add standard deviation error bars. Also, is the increase in heart rate at O1 and O2 in both groups due to pain? Please consider.

Yes, it corresponds to the average values, and we have included the standard deviation error bars as you requested.

Concerning the HR increase at O1 and O2, in our opinion, the end of the effect of the dexmedetomidine premedication at that time point could have contributed to this finding.

Thank you again for your accurate observations.

Round 2

Reviewer 3 Report

The submitted manuscript improved most of the points made. However, there is no discussion using the cited literature demonstrating that TAP blocks are effective for visceral pain. 

A comparative study by Cavaco, et al. applying TAP block to ovariectomy reported that TAP block was ineffective for intraoperative analgesia. The paper cited by the authors (Freitag, et al.) is a case report of only three cases using TAP blockade for visceral pain caused by pancreatitis, and is not a comparative study demonstrating that TAP blockade is effective for visceral pain. I believe that TAP block is an analgesic method that blocks somatic pain, unless there is a study with a higher level of evidence than that TAP block is effective for visceral pain (especially ovariectomy). In the authors' study, the control group did not receive saline at the TAP block site. Is it possible that the bupivacaine administered in the TAP blocker group was absorbed into the systemic circulation and exerted its analgesic effect? In any case, you should reconsider your results and discussions that TAP blocks are useful for postoperative analgesia in “laparoscopic ovariectomy,” as reported by Cavaco et al. (not laparoscopic surgery) recommend.

・Freitag FA, Bozak VL, do Carmo MP, Froes TR, Duque JC. Continuous transversus abdominis plane block for analgesia in three dogs with abdominal pain. Vet Anaesth Analg. 2018;45(4):581-583.

・Cavaco JS, Otero PE, Ambrósio AM, et al. Analgesic efficacy of ultrasound-guided transversus abdominis plane block in dogs undergoing ovariectomy. Front Vet Sci. 2022;9:1031345.

Author Response

The submitted manuscript improved most of the points made. However, there is no discussion using the cited literature demonstrating that TAP blocks are effective for visceral pain. 

A comparative study by Cavaco, et al. applying TAP block to ovariectomy reported that TAP block was ineffective for intraoperative analgesia. The paper cited by the authors (Freitag, et al.) is a case report of only three cases using TAP blockade for visceral pain caused by pancreatitis, and is not a comparative study demonstrating that TAP blockade is effective for visceral pain. I believe that TAP block is an analgesic method that blocks somatic pain, unless there is a study with a higher level of evidence than that TAP block is effective for visceral pain (especially ovariectomy). In the authors' study, the control group did not receive saline at the TAP block site. Is it possible that the bupivacaine administered in the TAP blocker group was absorbed into the systemic circulation and exerted its analgesic effect? In any case, you should reconsider your results and discussions that TAP blocks are useful for postoperative analgesia in “laparoscopic ovariectomy,” as reported by Cavaco et al. (not laparoscopic surgery) recommend. 

・Freitag FA, Bozak VL, do Carmo MP, Froes TR, Duque JC. Continuous transversus abdominis plane block for analgesia in three dogs with abdominal pain. Vet Anaesth Analg. 2018;45(4):581-583.

・Cavaco JS, Otero PE, Ambrósio AM, et al. Analgesic efficacy of ultrasound-guided transversus abdominis plane block in dogs undergoing ovariectomy. Front Vet Sci. 2022;9:1031345.

Thank you for providing us with the reference for the interesting study of Cavaco et al. The information from that study has been added to part of the discussion concerning visceral analgesia and has been properly cited in the part about postoperative pain and in the rest of the manuscript. The bibliography and drafting have been rearranged to include this citation.

In addition, we have replaced the term “perioperative analgesia” in the summary and text with “postoperative analgesia”.

Concerning your question about the possibility that bupivacaine was absorbed and exerted its analgesic effect systemically, we agree.

Petersen et al. also suggested this possibility for humans in a topical review published in 2010. According to the authors, post-operative pain can be considered a combination of both somatic (the abdominal wall incision) and visceral pain (internal organs), and the TAP block is not known to block visceral afferents. Therefore, other mechanisms of action with the TAP block have to be considered. Therefore, the authors suggested that the analgesic effect in part might be caused by a systemic rather than a local analgesic effect of the local anesthetic.

Despite this information, and although serum bupivacaine after TAP block in cats has been reported, the clinical significance of its effect on visceral pain has not been demonstrated or even suggested for cats, and serum bupivacaine concentration after TAP in dogs has not been studied. Therefore, we did not include these points in the discussion.

Petersen, P. L., Mathiesen, O., Torup, H., & Dahl, J. B. (2010). The transversus abdominis plane block: a valuable option for postoperative analgesia? A topical review. Acta Anaesthesiologica Scandinavica54(5), 529-535.

Garbin, M., Benito, J., Ruel, H. L., Watanabe, R., Monteiro, B. P., Cagnardi, P., & Steagall, P. V. (2022). Pharmacokinetics of bupivacaine following administration by an ultrasound-guided.

Thank you again for the time employed in revising in manuscript and for your suggestions that have led to a great improvement of the text.